# CodePMP: Scalable Preference Model Pretraining for Large Language Model Reasoning

## Abstract

Large language models (LLMs) have made significant progress in natural language understanding and generation, driven by scalable pretraining and advanced finetuning. However, enhancing reasoning abilities in LLMs, particularly via reinforcement learning from human feedback (RLHF), remains challenging due to the scarcity of high-quality preference data, which is labor-intensive to annotate and crucial for reward model (RM) finetuning. To alleviate this issue, we introduce CodePMP, a scalable preference model pretraining (PMP) pipeline that utilizes a large corpus of synthesized code-preference pairs from publicly available high-quality source code. CodePMP improves RM finetuning efficiency by pretraining preference models on large-scale synthesized code-preference pairs. We evaluate CodePMP on mathematical reasoning tasks (GSM8K, MATH) and logical reasoning tasks (ReClor, LogiQA2.0), consistently showing significant improvements in reasoning performance of LLMs and highlighting the importance of scalable preference model pretraining for efficient reward modeling.

## 1 Introduction

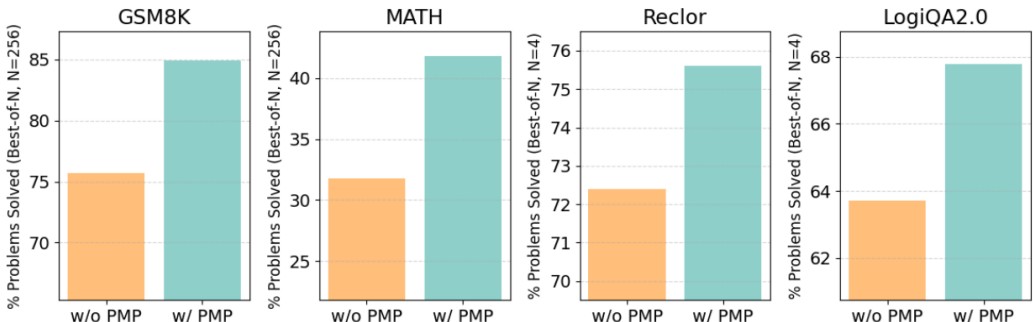

Figure 1: Compared to directly finetuning reward models, CodePMP significantly improves the sample efficiency and capability of reward models, which in turn boosts the generator's reasoning performance (Best-of-N accuracy) across both mathematical reasoning tasks (GSM8K and MATH) and logical reasoning tasks (ReClor and LogiQA2.0).

Large language models (LLMs) have made remarkable progress in natural language understanding and generation, benefiting from scalable pretraining and finetuning techniques like supervised finetuning (SFT) (Wang et al., 2022; 2023a) and Reinforcement Learning from Human Feedback (RLHF) (Bai et al., 2022a; Lightman et al., 2023b; Bai et al., 2022b; Gulcehre et al., 2023; Schulman et al., 2017; Rafailov et al., 2024). However, enhancing LLMs' reasoning abilities, particularly in complex logical and mathematical tasks, remains a significant challenge (Wang et al., 2023b; Zhang et al., 2024b). Although RLHF is effective, it relies heavily on high-quality preference data, which is costly and labor-intensive to annotate (Cobbe et al., 2021b; Zheng et al., 2024). This limitation impedes the scalability of reward model (RM) finetuning, which is essential for guiding LLMs toward optimal outputs.

Figure 2: **An illustration of the CodePMP process.** First, raw code collected from GitHub is cleaned and summarized into code prompts (descriptions). Then, for each code prompt, a weak CodeLLM generates a *rejected* response, while a stronger CodeLLM produces a *chosen* response. Finally, these *<chosen, rejected>* pairs, accumulated in the millions, form the pretraining dataset for the preference model. This pretraining process improves not only sample efficiency but also the performance for downstream reasoning reward model finetuning.

To alleviate this issue, prior works like Anthropic's Preference Model Pretraining (PMP) (Askell et al., 2021) have proposed improving reward modeling data efficiency by pretraining preference models on large-scale preference data from public sources like Reddit and Wikipedia, followed by an efficient finetuning on limited high-quality human-annotated data. However, this approach is less effective for reasoning tasks due to the scarcity of reasoning preference pairs available online. Compared to other tasks, manually annotating preference data for reasoning is inherently more challenging and difficult to scale (Zhang et al., 2024b; Zhou et al., 2023), highlighting the urgent need for a scalable PMP approach for reasoning tasks.

In this paper, we propose **CodePMP**, a scalable preference model pretraining pipeline that enhances LLM reasoning abilities using synthesized preference pairs derived from high-quality, publicly available source code. Code, with its inherently logical and structured nature, provides rich data suitable for reasoning tasks. Recent works (Zhang et al., 2024b; Aryabumi et al., 2024) also show a strong correlation between code training and reasoning improvements in LLMs. By leveraging the huge amount and diverse coverage of source code available on platforms like GitHub, CodePMP offers a scalable solution for pretraining preference models, thereby improving RM finetuning efficiency and enhancing LLMs reasoning performance.

Specifically, CodePMP generates preference pairs by synthesizing *chosen* and *rejected* code responses for a given code-related prompt or description using CodeLLMs. A strong CodeLLM produces higher-quality (*chosen*) responses, while a weaker model generates suboptimal or even low-quality (*rejected*) responses. These *<chosen, rejected>* pairs, accumulated in the millions, form a large-scale synthesized preference dataset. This dataset is then used to pretrain the preference model with pairwise ranking objectives (Cobbe et al., 2021b; Charniak & Johnson, 2005), providing an good initialization for further fine-tuning the reward models.

We evaluate CodePMP on widely studied reasoning tasks, including mathematical reasoning tasks such as GSM8K (Cobbe et al., 2021b) and MATH (Hendrycks et al., 2021), as well as logical reasoning tasks like ReClor (Yu et al., 2020) and LogiQA2.0 (Liu et al., 2023). Our experiments demonstrate that CodePMP significantly improves RM fine-tuning accuracy and Best-of-N performance in reasoning tasks, outperforming direct RM fine-tuning, as highlighted in Figure 1. Moreover, additional experimental results reveals that RMs initialized with CodePMP exhibit greater robustness across different tasks. These results indicate that code-derived preference data provides a scalable, cost-effective solution for enhancing LLM reasoning capabilities while reducing reliance on extensive preference annotation, achieving more effective reward modeling for reasoning tasks.

In summary, our main contributions are:

1. We introduce CodePMP, a scalable method that uses code-derived preference pairs to pretrain preference models, improving sample efficiency and robustness for downstream RM finetuning.

2. We validate that CodePMP significantly improves performance on reasoning tasks, demonstrating that a scalable PMP process positively impacts LLM reasoning abilities.

3. We provide a detailed analysis of key design elements in CodePMP, offering valuable insights for future research in related areas.

## 2 PRELIMINARIES

**Language Modeling**    Language modeling (LM) is a fundamental task in natural language processing, aimed at modeling sequences of language. This is typically achieved through Causal Language Models (Causal LM), where the model is trained to maximize the likelihood of predicting the next word $w_t$ given the preceding words $w_1, w_2, \ldots, w_{t-1}$. The training process minimizes the negative log-likelihood of the predicted word sequence:

$$\mathcal{L}_{\text{LM}} = -\sum_{t=1}^{T} \log P(w_t | w_1, w_2, \ldots, w_{t-1})$$

This loss function $\mathcal{L}_{\text{LM}}$ encourages the model to capture the underlying patterns in the data. Transformer architectures (Vaswani, 2017) have become the standard for Causal LM due to their ability to handle long-range dependencies effectively.

**Reward Modeling**    Reward modeling (RM) is crucial in reinforcement learning from human feedback (RLHF), providing a scalar reward signal that guides the learning process based on output quality. The reward model $R_\theta$ predicts the quality of an output $y$ given a context $x$ as $s = R_\theta(x, y)$, where $s$ is the scalar reward score. In preference modeling, RM predicts the relative quality of outputs by comparing pairs. A common method is the Pairwise Ranking Loss, where the model assigns higher scores to preferred (chosen) outputs:

$$\mathcal{L}_{\text{RM}} = -\log\left(\sigma(s_{\text{chosen}} - s_{\text{rejected}})\right)$$

, where $s_{\text{chosen}} = R_\theta(x, y_{\text{chosen}})$ and $s_{\text{rejected}} = R_\theta(x, y_{\text{rejected}})$, and $\sigma(\cdot)$ is the sigmoid function.

**Best-of-N Sampling**    Best-of-N (BoN) sampling improves LLM reasoning (Cobbe et al., 2021b; Lightman et al., 2023b). In this approach, $N$ candidate solutions $\{y_1, y_2, \ldots, y_N\}$ are generated by sampling from the LLM's output distribution for a given problem. A reward model scores each candidate and selects the highest-scoring one as the final answer:

$$\hat{y} = \arg\max_{y_i \in \{y_1, y_2, \ldots, y_N\}} R_\theta(x, y_i)$$

, where $R_\theta(x, y_i)$ represents the reward score for each candidate $y_i$. This technique is especially effective in tasks like mathematical problem-solving and logical inference, where selecting the most plausible solution from a diverse set of outputs improves overall accuracy (Wang et al., 2022).

## 3 CODE PREFERENCE MODEL PRETRAINING

### 3.1 MODEL DESIGN

Code Preference Model Pretraining (CodePMP) is designed to enhance the sample efficiency of reward models, particularly for reasoning tasks where high-quality preference data is scarce. Traditionally, reward models are finetuned on small, curated datasets, which limits their effectiveness in complex tasks like mathematical reasoning or logical deduction. CodePMP mitigates this limitation by introducing a pretraining phase between basic language model pretraining and finetuning on domain-specific reasoning datasets. This phase leverages a large, diverse dataset of code-preference pairs, enabling the model to learn generalizable patterns and ranking strategies.

CodePMP training involves two key components: Reward Modeling (RM) and Language Modeling (LM). In RM, the model is trained on code-preference pairs, learning to assign higher scores to the *chosen* code through a pairwise ranking loss. In LM, only the *chosen* code is used for autoregressive

---

**Algorithm 1** Code Preference Model Pretraining (CodePMP)

---

**Require:** Source code repository $S$, Strong CodeLLM $M_{\text{strong}}$, Weak CodeLLM $M_{\text{weak}}$
**Ensure:** Pretrained Model
  1: **Input:** Source code $S$
  2: Summarize description $D$ using $M_{\text{strong}}$ on $S$
  3: **for** each $D_i \in D$ **do**
  4:     Generate *Chosen Response* using $M_{\text{strong}}$
  5:     Generate *Rejected Response* using $M_{\text{weak}}$
  6: **end for**
  7: Calculate LM Loss $\mathcal{L}_{\text{LM}}$ on *Response*
  8: Calculate RM Loss $\mathcal{L}_{\text{rank}}$ using *Chosen Response* and *Rejected Response*
  9: Train PMP Model using $\mathcal{L}_{\text{PMP}} = \mathcal{L}_{\text{rank}} + \mathcal{L}_{\text{LM}}$

---

training to maintain the model's general capabilities. The overall loss is a combination of the RM and LM losses, ensuring the model enhances its ranking ability without sacrificing general language modeling performance: $\mathcal{L}_{\text{PMP}} = \mathcal{L}_{\text{rank}} + \mathcal{L}_{\text{LM}}$.

### 3.2 DATA CONSTRUCTION

To enable scalable preference model pretraining, we construct a dataset sourced from GitHub, which includes a diverse range of repositories and associated metadata. The dataset consists of two primary components: **Repository Data** comprises over 1.3 billion code files from GitHub repositories, while **GitHub Metadata** includes information such as commit histories, discussions, pull requests, and issues.

The CodePMP dataset is constructed through a systematic process. First, raw source code is processed by a description summarizer, typically an instruction-tuned CodeLLM, to generate prompts that describe the functionality of the code.

These prompts are then used by two CodeLLMs of different capabilities to generate code snippets:

- **Chosen response**: Generated by a more advanced CodeLLM (e.g., 6.7B parameters).
- **Rejected response**: Generated by a less capable CodeLLM (e.g., 1.3B parameters).

This process yields pairs of code responses—one chosen and one rejected—which are used for preference modeling. This scalable approach significantly enhances pretraining efficiency, improving performance on downstream tasks.

The steps of the CodePMP methodology are outlined systematically in Algorithm 1.

## 4 EXPERIMENTAL

In this section, we first outline the experimental setup, followed by the experimental results, highlighting that CodePMP is a highly scalable method.

### 4.1 EXPERIMENTAL SETTINGS

#### 4.1.1 CODEPMP SETTINGS

**Data Construction**   We generate code preference pairs following Algorithm 1, using the deepseek-coder-6.7b-instruct model as the strong CodeLLM to generate *chosen* responses and the deepseek-coder-1.3b-instruct model as the weak CodeLLM to generate *rejected* responses. The constructed CodePMP dataset includes 28 million files and 19 billion tokens. The diverse datasets provide sufficiently broad prompt coverage for preference model pretraining, which is conducive to the generalization of preference models in reasoning tasks. In addition, the average lengths of the *chosen* and *rejected* responses are similar, ensuring that response length does not bias the CodePMP learning process. Details are provided in Appendix C and Table 4.

**CodePMP Training** By default, we initialize the preference models with the publicly available Qwen models (Yang et al., 2024a), using different model sizes, specifically Qwen2-1.5B and Qwen2-7B. Detailed hyperparameters for CodePMP training are provided in Appendix B.

### 4.1.2 REASONING FINETUNING SETTINGS

We validate CodePMP on reward models across two reasoning task types: mathematical and logical reasoning. The reward model is finetuned on corresponding preference datasets for each task. For mathematical reasoning, we use the MathShepherd-pair dataset, derived from MathShepherd (Wang et al., 2023b), and evaluate the model on a holdout test set to assess RM accuracy.

Similarly, for logical reasoning, we use the ReClor-pair and LogiQA2.0-pair datasets, derived from ReClor (Yu et al., 2020) and LogiQA2.0 (Liu et al., 2023), respectively. We train reward models on these datasets, with holdout test sets used to evaluate model accuracy. Dataset construction and finetuning hyperparameters are provided in Appendix D and B.

### 4.1.3 EVALUATION SETTINGS

Following (Zhang et al., 2024a), we adopt two evaluation metrics:

**RM Accuracy** This metric measures the accuracy of the reward model in distinguishing chosen from rejected solutions on the holdout test sets. It provides insight into the model's ability to classify individual sequences.

**Best-of-N (BoN) Accuracy** This metric evaluates the proportion of correct solutions selected by the finetuned RM from $N$ candidate responses. It assesses the model's group-wise ranking performance, focusing on its ability to select the correct answer from a set of candidates. We use MetaMath-Mistral-7B (Yu et al., 2023) as the generator for this evaluation.

For mathematical reasoning, we use the GSM8K (Cobbe et al., 2021b) and MATH (Hendrycks et al., 2021) test sets. For logical reasoning, we evaluate on the ReClor (Yu et al., 2020) and LogiQA2.0 (Liu et al., 2023) test sets. Further details can be found in Appendix D.

Note that logical reasoning questions typically involve a paragraph followed by statements to be judged true or false, making Best-of-N evaluation challenging. Therefore, we use multiple-choice accuracy, where the reward model ranks four manually annotated options and selects the best one. This metric is equivalent to Best-of-4, and thus, for logical reasoning tasks, multiple-choice accuracy and Best-of-N are used interchangeably.

### 4.2 EXPERIMENTAL RESULTS

#### 4.2.1 RM ACCURACY RESULTS

We first compare RM accuracy on the holdout test set with and without CodePMP initialization. As shown in Table 1, RM finetuned with CodePMP initialization achieves higher accuracy on both 1.5B and 7B models across mathematical and logical reasoning tasks, demonstrating that CodePMP enhances the model's ability to differentiate correct from incorrect reasoning. Moreover, CodePMP exhibits strong generalization, yielding significant improvements across different reasoning tasks.

#### 4.2.2 BON ACCURACY RESULTS

We evaluate BoN accuracy across reasoning tasks with and without CodePMP initialization. As shown in Figure 8a and Table 8b, RM finetuned with CodePMP initialization consistently achieves higher BoN accuracy across both mathematical and logical reasoning tasks for 1.5B and 7B models. This highlights CodePMP's effectiveness in improving RM's group-wise ranking performance.

Across different values of N, RM models initialized with CodePMP maintain their lead, showing robust improvement even as N increases to 256. In contrast, RM without CodePMP shows a sharp decline in accuracy as N increases, underscoring the stability CodePMP provides, likely due to the diverse code-preference pairs used during training.

Table 1: Comparison of RM accuracies: reward models finetuned with CodePMP initialization achieve higher accuracies on the reasoning holdout test sets, demonstrating an improved ability to distinguish chosen responses from rejected ones.

| Model | PMP | MathShepherd-pair | Reclor-pair | LogiQA2.0-pair |
|-------|-----|-------------------|-------------|----------------|
| 1.5B  | ✗   | 0.7226            | 0.758       | 0.7538         |
|       | ✓   | **0.8186**        | **0.794**   | **0.7774**     |
| 7B    | ✗   | 0.8777            | 0.862       | 0.8263         |
|       | ✓   | **0.9274**        | **0.874**   | **0.8441**     |

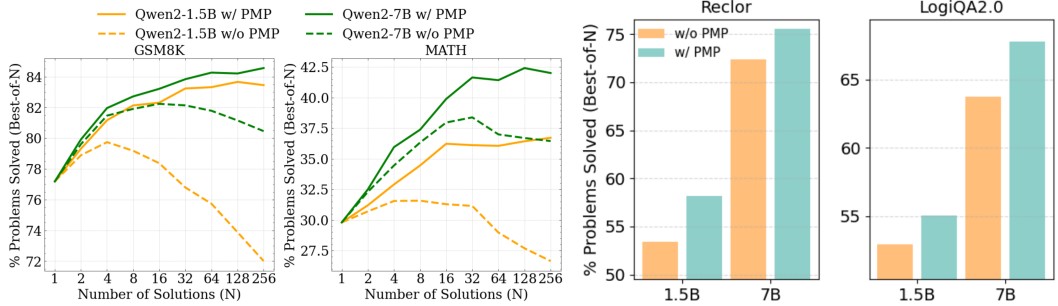

(a) BoN accuracies on mathematical reasoning.   (b) BoN *(N=4)* accuracies on logical reasoning.

Figure 3: Comparison of Best-of-N accuracies: reward models finetuned with CodePMP initialization consistently perform better, with improvements remaining robust as N increases, demonstrating CodePMP's effectiveness in improving group-wise ranking capabilities.

For logical reasoning, the performance gap between CodePMP and non-PMP models is smaller, as $N$ is limited to 4, while in mathematical reasoning, $N$ reaches 256. This suggests that increasing $N$ further in logical reasoning could amplify the advantages of CodePMP in future evaluations.

### 4.2.3 RM SAMPLE EFFICIENCY COMPARISON

One key advantage of CodePMP is its ability to improve the sample efficiency of RM finetuning. To assess this, we conduct experiments with progressively larger sample sizes for RM finetuning. As indicated by (Kaplan et al., 2020), optimal results are achieved when the learning rate scheduler completes its decay at the end of training. Therefore, rather than evaluating intermediate checkpoints, we retrain models with varying sample sizes for optimal results. Figure 4 and 10 (Appendix E) show that as the sample size increases, RMs with CodePMP initialization consistently outperforms others in both BoN and RM accuracy. Notably, RMs finetuned with CodePMP initialization using just 0.5k samples surpasses RMs finetuned without CodePMP initialization using 40k samples on mathematical tasks, demonstrating CodePMP's significant advantage in sample efficiency. However, as sample size increases, this advantage diminishes slightly, suggesting that with much larger datasets, CodePMP's benefit may become less pronounced, but the cost of manual labeling remains a key consideration.

### 4.2.4 THE IMPORTANCE OF SCALABLE PMP

A key benefit of using code data for PMP is the vast availability of publicly accessible, high-quality code-preference pairs, ensuring diversity. To validate scalability, we vary the number of training pairs for CodePMP and retrain models with different amounts of data. As shown in Figure 5, overall, increasing the number of code-preference pairs consistently improves BoN accuracy in both mathematical and logical reasoning tasks across model sizes, with no sign of diminishing returns. This indicates that further scaling the code-preference data would likely yield additional performance gains, underscoring the importance of building a scalable PMP pipeline.

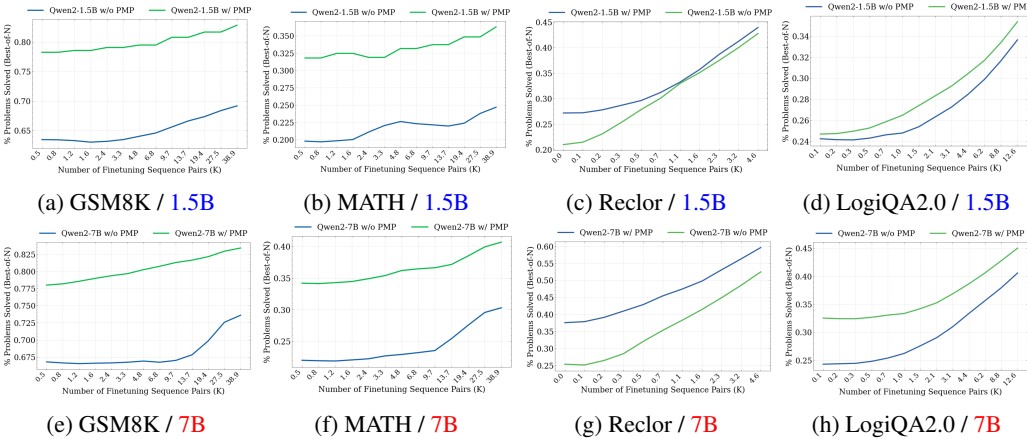

(a) GSM8K / 1.5B        (b) MATH / 1.5B        (c) Reclor / 1.5B        (d) LogiQA2.0 / 1.5B

(e) GSM8K / 7B        (f) MATH / 7B        (g) Reclor / 7B        (h) LogiQA2.0 / 7B

Figure 4: Comparison of sample efficiency in RM finetuning: reward models finetuned with CodePMP initialization consistently achieve substantially higher Best-of-N accuracy when finetuning on the same amount of samples, demonstrating superior sample efficiency. Note that the horizontal axis grows exponentially with $\sqrt{2}$. The green lines represent the settings with CodePMP, while the blue lines represent the settings without CodePMP.

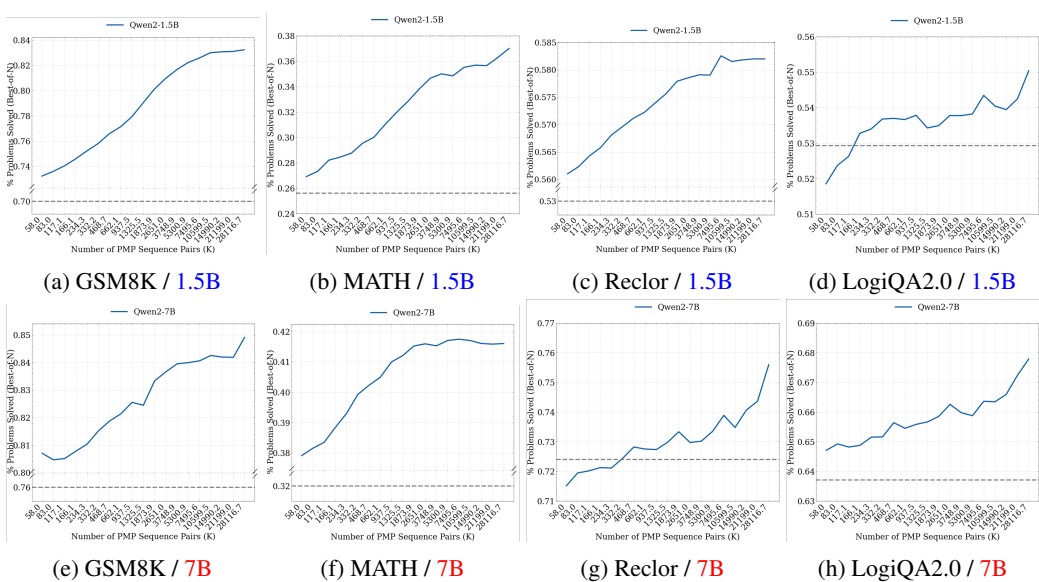

(a) GSM8K / 1.5B        (b) MATH / 1.5B        (c) Reclor / 1.5B        (d) LogiQA2.0 / 1.5B

(e) GSM8K / 7B        (f) MATH / 7B        (g) Reclor / 7B        (h) LogiQA2.0 / 7B

Figure 5: Increasing the number of code-preference pairs consistently improves Best-of-N accuracy in both mathematical and logical reasoning tasks across model sizes, with no evident signs of diminishing returns. Note that the horizontal axis is scaled by $\sqrt{2}$, and the gray dashed line represents the results without CodePMP.

## 5 ABLATION STUDIES

In this section, we present a detailed analysis of CodePMP design. Unless otherwise stated, all experiments used the 1B model due to resource limitations and present the results of mathematical reasoning due to page limitation. Results of logical reasoning refers to Appendix E.2.

### 5.1 IMPACT OF PAIR CONSTRUCTION

**Model-Generated Pairs Comparison** We compare various pair construction methods generated by different models. In Figure 6a, the samples before the "" are positive, and those after are negative.

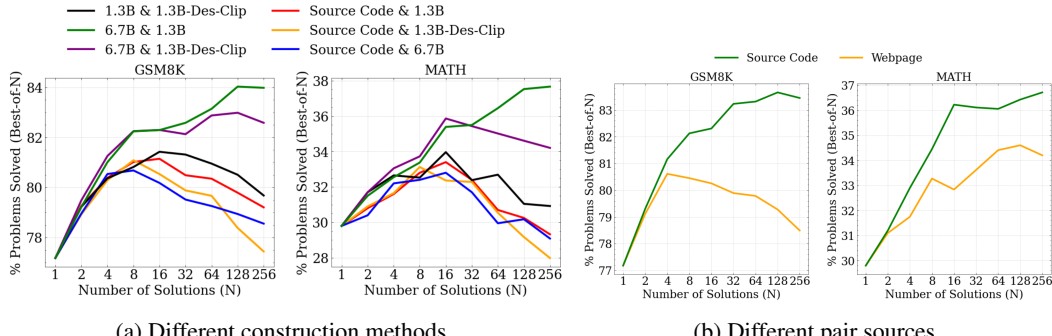

(a) Different construction methods.

(b) Different pair sources.

Figure 6: Comparisons of BoN accuracy across different construction methods and pair sources.

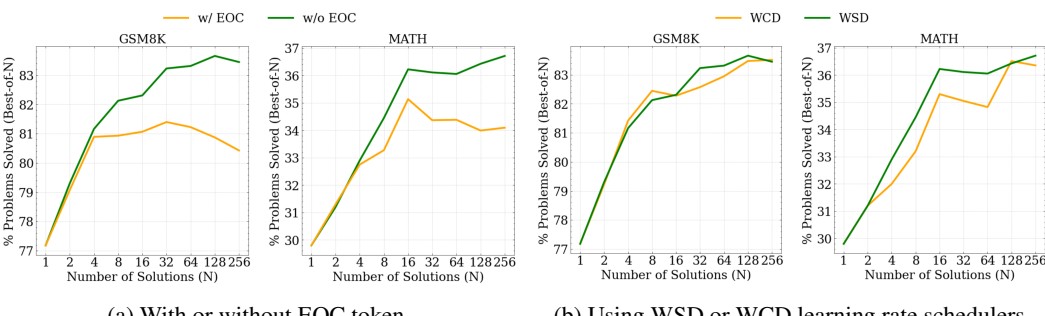

(a) With or without EOC token.

(b) Using WSD or WCD learning rate schedulers.

Figure 7: Comparisons of BoN accuracy for different settings: EOC token and lr schedulers.

"Source Code" refers to the original code snippet, while "1.3B-Des-Clip" indicates that 10% of the code description is removed before being input into a 1.3B CodeLLM to generate a rejected response. The green lines represent CodePMP's choice. Results show that pairing positive samples from the 7B model with negative samples from the 1.5B model consistently delivers the best performance across all test sets. Given that code execution can generate reliable outputs, future work will explore incorporating execution feedback to create more accurate preference pairs.

**Web-Crawled vs GitHub-Sourced Pairs** We also compare GitHub-sourced code with web-crawled code data( Askell et al. (2021)) from platforms such as StackExchange and Reddit. As shown in Figure 7a, GitHub-sourced pairs ("Source Code") consistently outperform those from web platforms ("Webpage"), particularly as the number of solutions (N) increases. Moreover, the performance improvement of GitHub-sourced pairs shows no sign of plateauing, highlighting the importance of diverse, high-quality source code in building a scalable PMP pipeline.

## 5.2 IMPACT OF EOC TOKEN

Experiments by( Askell et al. (2021)) show that adding an end-of-context (EOC) token to each sequence significantly improves overall performance. To explore its impact in the context of CodePMP, we compared performance with and without the EOC token. As shown in Figure 7a, the EOC setting ("w/ EOC") consistently underperform the setting without EOC tokens ("w/o EOC") across different test tasks, which is opposed to ( Askell et al. (2021)) We attribute this discrepancy to the different model, data and evaluation settings.

## 5.3 IMPACT OF LEARNING RATE SCHEDULERS

In the CodePMP experiments, we use the warmup-stable-decay (WSD) learning rate scheduler( Hu et al. (2024)), which can effectively reduce the time required for scaling related experiments. Previous studies mainly employ a learning rate schedule with linear warmup followed by cosine decay, known as warmup-cosine decay (WCD). We compare the performance of WSD and WCD, as shown in

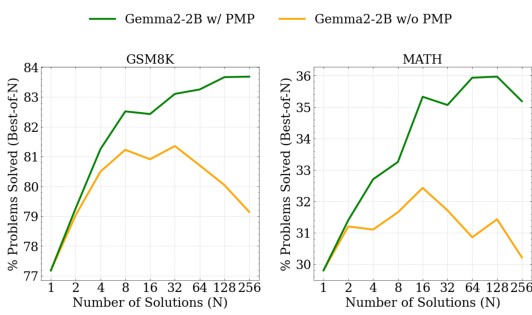
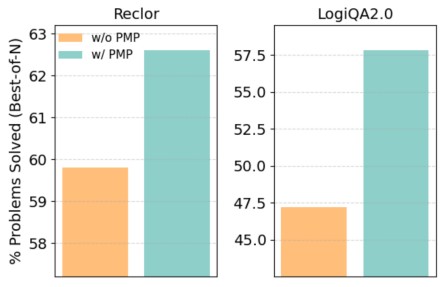

(a) BoN accuracies on mathematical reasoning.  (b) BoN *(N=4)* accuracies on logical reasoning.

Figure 8: CodePMP improves sample efficiency and Best-of-N performance of the Gemma-2B reward models on reasoning tasks, highlighting its broad applicability across diverse LLM architectures.

Table 2: Performance gains on coding RM and general RM (RMBench) evaluations show that CodePMP not only improves reasoning tasks but also generalizes well across various tasks.

| Model | PMP | Coding | Summary | Chat | RMBench Chat Hard | Safety | Reasoning |
|-------|-----|--------|---------|------|-------------------|--------|-----------|
| **1.5B** | ✗ | 0.6841 | 0.4154 | 0.4804 | **0.5351** | 0.3665 | 0.2751 |
|  | ✓ | **0.758** | **0.6126** | **0.9050** | 0.4364 | **0.3698** | **0.6041** |
| **7B** | ✗ | 0.6912 | 0.5839 | 0.4972 | 0.5022 | **0.5240** | 0.6804 |
|  | ✓ | **0.7619** | **0.7668** | **0.9413** | **0.5373** | 0.4906 | **0.9116** |

Table 7b, both schedulers yield similar results. Thus, to improve computational efficiency, we adopt the WSD scheduler for all experiments.

## 5.4 VALIDATING CODEPMP ON OTHER LLMS

To further evaluate the generalizability of CodePMP, we validate its performance on the widely adopted Gemma-2B model (Team et al., 2024). As illustrated in Figure 8, the application of CodePMP results in significant performance gains in both mathematical reasoning and logical reasoning evaluations. This not only underscores the robustness of CodePMP but also demonstrates its broad applicability in improving sample efficiency and overall performance across diverse LLM architectures.

## 5.5 PERFORMANCE ON CODING AND GENERAL RM BENCHMARKS

We evaluate CodePMP on both code-specific and general reward modeling benchmarks. The CodeUltraFeedback_binarized test set serves as an in-domain evaluation, while RMBench provides an out-of-domain assessment. As shown in Table 2, models finetuned with CodePMP initialization consistently outperform those without CodePMP across various model sizes. These results demonstrate that CodePMP not only enhances performance in reasoning tasks but also generalizes well across a range of RM benchmarks.

## 6 RELATED WORKS

**Reward Modeling** Reward models (RMs) in RLHF have traditionally used ranking models like Bradley-Terry and Plackett-Luce to capture human preferences (Bradley & Terry, 1952; Plackett, 1975; Cobbe et al., 2021b; Saunders et al., 2022; Lightman et al., 2023b; Wang et al., 2023b; Uesato et al., 2022; Luo et al., 2024; Yu et al., 2024; Stiennon et al., 2020; Nakano et al., 2021). Recent advancements introduced probability-based methods (Zhao et al., 2023; Jiang et al., 2023), offering more refined predictions. Innovations such as the Critique-out-Loud model (Ankner et al., 2024)

integrate natural language critiques to enhance RMs. Generative reward models (GRMs) (Yang et al., 2024b) further improve sample efficiency. Preference Modeling Pretraining (PMP) (Askell et al., 2021) introduces a pretraining phase, leveraging large-scale pairwise ranking data to boost RM performance. However, many of these methods rely on costly manual annotations or limited data, limiting scalability. CodePMP addresses this issue by automating preference data generation from code, improving RM sample efficiency and reducing dependency on manual data collection.

**Code Training** Incorporating code into LLM pretraining has significantly improved performance in tasks such as commonsense reasoning (Madaan et al., 2022) and mathematical reasoning (Liang et al., 2022; Shao et al., 2024; Yang et al., 2024a). Code also enhances general reasoning abilities (Muennighoff et al., 2023; Fu & Khot, 2022; Ma et al., 2023). Recent research (Dong et al., 2023; Ma et al., 2023) shows that integrating code during supervised finetuning strengthens LLMs in complex decision-making tasks. CodePMP pioneers the use of scalable, synthetically generated code preference pairs, reducing reliance on manual annotations (Dubey et al., 2024; Gemini-Team et al., 2024; Groeneveld et al., 2024; Bi et al., 2024). This approach improves sample efficiency and scalability in reasoning-intensive tasks, opening new possibilities for LLM performance improvements.

**LLM Reasoning** Improving reasoning in LLMs remains a challenge, and various advanced techniques have been proposed. Chain of Thought (CoT) prompting (Wei et al., 2022; Fu et al., 2023) improves reasoning by generating intermediate steps, while supervised finetuning (SFT) with CoT further boosts performance (Cobbe et al., 2021a; Liu et al., 2024; Yu et al., 2023). Other methods focus on increasing inference time computation, such as problem decomposition (Zhou et al., 2022), search-based approaches like MCTS (Xu, 2023), and using LLMs as verifiers (Huang et al., 2022; Luo et al., 2023). Reward models, including outcome-based (ORM) and process-based (PRM), also improve performance, with PRM showing stronger results (Lightman et al., 2023a; Wang et al., 2023b). Unlike these approaches, CodePMP introduces a scalable preference model pretraining stage, which is compatible with all the aforementioned methods.

# 7 CONCLUSION & FUTURE WORKS

This paper introduces **CodePMP**, a scalable pretraining approach that leverages code-preference pairs to improve reasoning capabilities in large language models. Experimental results validate that CodePMP significantly improves sample efficiency and boosts performance on reasoning tasks.

For future work, we aim to extend CodePMP in two key directions. **CodePrMP** will focus on utilizing compiler and interpreter verifiability to provide low-cost process supervision signals. **GenPMP** will explore how to improve sample efficiency and the performance of generative reward models by integrating code data.

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

## A  HYPERPARAMETERS

We outline key hyperparameters in this Table 3. In the tables, WSD refers to the warmup-stable-decay learning rate scheduler( (Hu et al., 2024)), which has the benefit of reducing the time required for scaling law experiments.

Table 3: Hyperparameters for CodePMP training, mathematical reasoning RM finetuning, and logical reasoning RM finetuning.

| HP | CodePMP | | Mathematical RM | | Logical RM | |
|---|---|---|---|---|---|---|
| | 1.5B | 7B | 1.5B | 7B | 1.5B | 7B |
| epoch | 1 | 1 | 1 | 1 | 1 | 1 |
| bs | 1024 | 1024 | 64 | 64 | 64 | 64 |
| lr | 3e-6 | 1e-6 | 1e-6 | 3e-7 | 1e-5 | 1e-5 |
| lr scheduler | WSD | WSD | WCD | WCD | WCD | WCD |
| warmup ratio | 0.03 | 0.03 | 0.03 | 0.03 | 0.25 | 0.25 |
| decay ratio | 0.1 | 0.1 | - | - | - | - |
| weight decay | 0.1 | 0.1 | 0 | 0 | 0 | 0 |
| max length | 1024 | 1024 | 1024 | 1024 | 1024 | 1024 |

## B  TRAINING PIPELINE

Figure B presents an overview of the complete training pipeline. The process begins with base language model (LM) pretraining on trillions of tokens from general text, followed by a preference model pretraining (PMP) phase using billions of tokens from code preference pairs. Finally, the model is finetuned on a smaller, more specialized dataset relevant to reasoning tasks, typically consisting of millions of tokens.

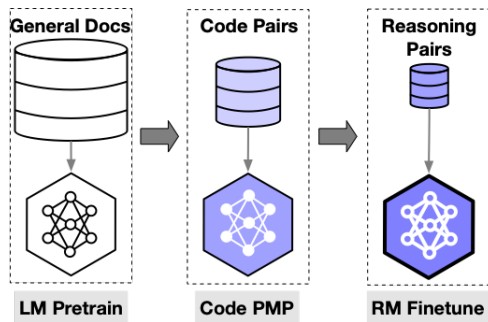

Figure 9: An overview of the complete training pipeline.

## C  CODEPMP DATASET

As shown in Table 4, the constructed CodePMP dataset consists of a total of 28 million files, accounting for 19 billion tokens across various languages. The dataset is primarily composed of Python files, with 20 million files and 13.1 billion tokens, followed by Notebook files, contributing 3 million files and 2.1 billion tokens, and other programming languages with 5 million files and 3.8 billion tokens. This diverse dataset supports the pretraining phase, aiding the model in generalizing across multiple reasoning tasks.

The average length of *chosen* responses varies slightly between different languages. Python files exhibit an average response length of 170 tokens for chosen responses and 167 tokens for rejected responses, while Notebook files have slightly shorter average lengths of 158 and 155.5 tokens, respectively. Other languages show the highest average response lengths, with 213.2 tokens for

Table 4: Amount and average length statistics of CodePMP dataset.

| Language | Amount Statistics | | Average Length | |
| | Files (M) | Tokens (B) | Chosen | Rejected |
| --- | --- | --- | --- | --- |
| Python | 20 | 13.1 | 170.0 | 167.0 |
| Notebook | 3 | 2.1 | 158.0 | 155.5 |
| Other Languages | 5 | 3.8 | 213.2 | 210.0 |
| **Total** | **28** | **19.0** | **194.5** | **189.9** |

chosen and 210 tokens for rejected responses. Overall, the average length of chosen responses across the dataset is 194.5 tokens, while the average length of rejected responses is 189.9 tokens, indicating minimal bias in the dataset based on response length.

## D   RM FINETUNING DATASET

### D.1   MATHEMATICAL REASONING

The RM finetuning for mathematical reasoning uses the MathShepherd dataset( (Wang et al., 2023b)), which contains 444k query-response samples, with some queries having multiple distinct responses. We divide the dataset into a 400k training set and a 44k test set. For RM finetuning, we construct preference pairs by selecting both correct and incorrect responses for the same query. To form the 4.3k test set, we combine one positive and negative sample for each query from the original test set.

We also create two training sets of different sizes: MathShepherd-preference-800k and MathShepherd-preference-40k. The 800k training set is built by combining multiple positive and negative samples for each query in the original training set, resulting in 800k samples. In contrast, the 40k training set randomly selects one positive-negative pair for each query, totaling 40k samples.

### D.2   LOGICAL REASONING

#### D.2.1   RECLOR

Reclor is a human-annotated reading comprehension reasoning dataset, where each sample consists of a passage, a question, and multiple options. To create preference pairs, we combine the correct and incorrect options for the same question. This process results in a total of 14.5k preference pairs, with 14k pairs used for training and 1.5k for testing, forming the Reclor-preference dataset.

#### D.2.2   LOGIQA2.0

For logical reasoning finetuning scaling analysis, we synthesize a preference dataset based on the logical reasoning dataset LogiQA2.0. LogiQA2.0 is a reading comprehension benchmark requiring discrete reasoning over passages, with 96k crowdsourced, adversarially created questions. To answer correctly, models must resolve references (which may point to multiple locations in the input) and perform discrete operations like addition, counting, or sorting. We use four models (Qwen2-7B-Instruct[1], Qwen2-72B-Instruct[2], DeepSeek-V2-Chat[3], and DeepSeek-Coder-V2-Instruct[4]) to sample queries from the LogiQA2.0 dataset multiple times, with sampling $topp = 1$, and $topk \in \{0, 0.01, 0.1, 0.2, 0.3, 0.4, 0.5, 0.6, 0.7, 0.8, 0.9, 1.0\}$. Answer correctness is annotated using DeepSeek-7B-Math-Compare-Answer[5]. Correct and incorrect answers are combined to create preference pairs, resulting in 1,019k pairs, with 977k used for training and 42k for testing, forming the LogiQA2.0 preference dataset for RM scaling analysis.

---

[1]https://huggingface.co/Qwen/Qwen2-7B-Instruct

[2]https://huggingface.co/Qwen/Qwen2-72B-Instruct

[3]https://huggingface.co/deepseek-ai/DeepSeek-V2-Chat

[4]https://huggingface.co/deepseek-ai/DeepSeek-Coder-V2-Instruct

[5]https://huggingface.co/Tianqiao/DeepSeek-7B-Math-Compare-Answer

### D.3 CODEULTRAFEEDBACK_BINARIZED

CodeUltraFeedback_binarized[6] is a preference dataset in the code domain, consisting of 9.5k preference pairs. We randomly split the dataset, using 90% of the samples for finetuning training and 10% for testing RM accuracy.

## E MORE EXPERIMENTAL RESULTS

Due to the length of the paper, we only report important experimental results in the experiment and ablation sections. In this section, we present complementary results in more detail.

### E.1 RM SAMPLE EFFICIENCY COMPARISON: RM ACCURACY RESULTS

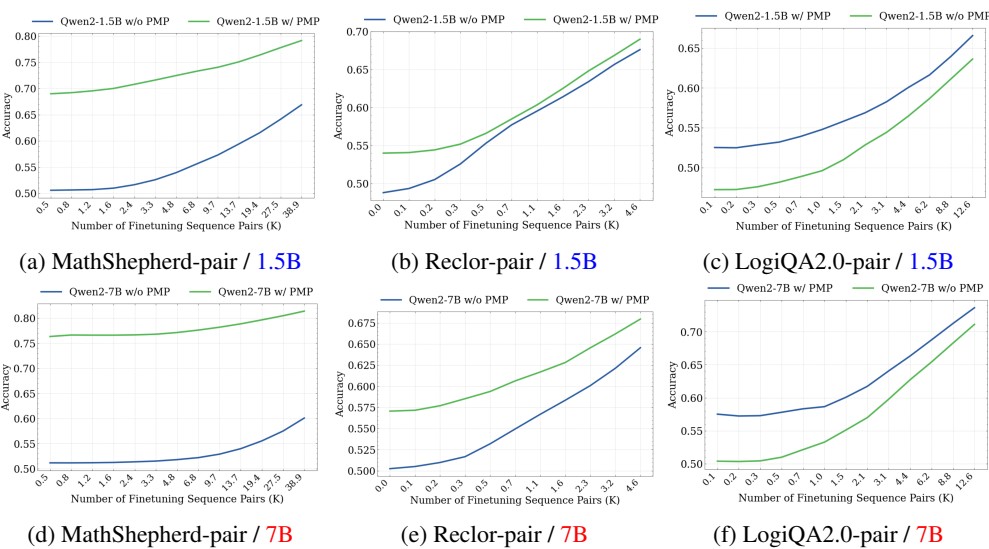

(a) MathShepherd-pair / 1.5B     (b) Reclor-pair / 1.5B     (c) LogiQA2.0-pair / 1.5B

(d) MathShepherd-pair / 7B     (e) Reclor-pair / 7B     (f) LogiQA2.0-pair / 7B

Figure 10: Comparison of sample efficiency of RM finetuning: Trends of RM accuracy with sample size increases. **Note:** The horizontal axis increases exponentially with $\sqrt{2}$. Therefore, the further the data intervals, the more RM data is effectively saved. We use different colors to highlight the results of different model sizes.

We use RM accuracy evaluation to compare the sample efficiency of RM finetuning, as shown in Figure 10. Consistent with the conclusion from the main experiment, CodePMP improves sample efficiency. Under the same sample conditions, it consistently delivers stable improvements, making RM training more efficient.

### E.2 ABLATION RESULTS ON LOGICAL REASONING

Table 5 shows the comparison of ReClor and LogiQA2.0 BoN accuracies for different Model-generated pair construction methods for CodePMP. Overall, the $6.7B\&1.3B$ setting is the best among them.

Table 6 shows the comparison of the BoN accuracy performance of ReClor and LogiQA2.0 using GitHub source code and web crawled code data to build CodePMP training pairs. The results verify that Github code is a better source for pair construction.

Table 7 shows the comparison of ReClor and LogiQA2.0 BoN accuracies for CodePMP models with and without EOC. In general, the setup without using EOC brings better results.

Table 8 shows the comparison of ReClor and LogiQA2.0 BoN accuracies for WCD and WSD lr scheduler settings. The results below show that WSD lr scheduler brings better results than WCD.

---

[6]https://huggingface.co/datasets/coseal/CodeUltraFeedback_binarized

Table 5: Comparison of ReClor and LogiQA2.0 BoN accuracies for different Model-generated pair construction methods for CodePMP. Overall, the $6.7B\&1.3B$ setting is the best among them.

| RM Finetune | Constructions Method | Test Set | |
|---|---|---|---|
| | | ReClor | LogiQA2.0 |
| ReClor-pair | Source Code & 1.3B | 0.550 | 0.4464 |
| | Source Code & 1.3B-Des-Clip | 0.600 | 0.4522 |
| | Source Code & 6.7B | 0.578 | 0.4605 |
| | 1.3B & 1.3B-Des-Clip | 0.572 | 0.4745 |
| | 6.7B & 1.3B-Des-Clip | 0.564 | 0.4809 |
| | 6.7B & 1.3B | **0.608** | **0.5032** |
| LogiQA2.0-pair | Source Code & 1.3B | 0.708 | 0.5217 |
| | Source Code & 1.3B-Des-Clip | 0.704 | 0.5268 |
| | Source Code & 6.7B | 0.714 | 0.5198 |
| | 1.3B & 1.3B-Des-Clip | 0.738 | 0.5402 |
| | 6.7B & 1.3B-Des-Clip | **0.742** | 0.5274 |
| | 6.7B & 1.3B | 0.734 | **0.5415** |

Table 6: Comparison of the BoN accuracy performance of ReClor and LogiQA2.0 using GitHub source code and web crawled code data to build CodePMP training pairs. The results below show that Github code is a better source for pair construction.

| RM Finetune | Data Source | Test Set | |
|---|---|---|---|
| | | ReClor | LogiQA2.0 |
| ReClor-pair | Webpage | 0.574 | 0.4898 |
| | Github | **0.582** | **0.4981** |
| LogiQA2.0-pair | Webpage | 0.742 | 0.5293 |
| | Github | **0.752** | **0.5504** |

### E.3 CROSS-VALIDATIONS ON LOGICAL REASONING

We further conduct cross-dataset validation on logical reasoning. Specifically, we evaluate the RM models finetuned on the ReClor-pairs dataset using the LogiQA2.0 test set. Similarly, we evaluate the RM models finetuned on the LogiQA2.0-pairs dataset using the ReClor test set. As shown in Table 9, CodePMP consistently improves RM evaluation performance, demonstrating that the enhancements CodePMP brings to RM training are robust and generalizable. Note that the BoN accuracies of the RM trained with LogiQA2.0-pair on the Reclor test set are higher than those of the RM trained directly on the Reclor-pair, because the LogiQA2.0-pair dataset is three times larger than the Reclor-pair dataset.

## F LOGICAL REASONING EVALUATION EXAMPLES

We randomly select and present examples from the Reclor test set, which consists of multiple-choice questions based on a given passage. While it is possible to have the model generate additional candidate answers to create a Best-of-N test, it becomes difficult to ensure that the original correct answer remains among the options after introducing new candidates, and to identify the new correct answer. We attempt to use GPT-4o to annotate the correct answers for 32 responses, but the consistency with manual inspection is low, as is the consistency of GPT-4o's own multiple annotations. It can be inferred that the consistency rate would worsen if expanded to 256 responses. Therefore, after careful consideration, we decide to use RM to score only the original four manually annotated answer options, match the top-ranked option with the manually annotated correct answer, and calculate accuracy. In principle, this method is equivalent to the Best-of-4 test.

Table 7: Comparison of ReClor and LogiQA2.0 BoN accuracies for CodePMP models with and without EOC. In general, the setup without using EOC brings better results.

| RM Finetune | Model | Test Set | |
| --- | --- | --- | --- |
| | | ReClor | LogiQA2.0 |
| ReClor-pair | w/o EOC | 0.582 | **0.4981** |
| | w/ EOC | **0.596** | 0.4617 |
| LogiQA2.0-pair | w/o EOC | **0.752** | **0.5504** |
| | w/ EOC | 0.686 | 0.4809 |

Table 8: Comparison of ReClor and LogiQA2.0 BoN accuracies for WCD and WSD lr scheduler settings. The results below show that WSD lr scheduler brings better results than WCD.

| RM Finetune | Model | Test Set | |
| --- | --- | --- | --- |
| | | ReClor | LogiQA2.0 |
| ReClor-pair | WCD | 0.552 | 0.4828 |
| | WSD | **0.582** | **0.4981** |
| LogiQA2.0-pair | WCD | 0.748 | 0.5204 |
| | WSD | **0.752** | **0.5504** |

Table 10: Examples from the Reclor test set, which consists of multiple-choice questions based on a given passage.

| ID | Text | Options | Answer |
| --- | --- | --- | --- |
| 12824 | Mayor: Four years ago, when we reorganized the city police department in order to save money, critics claimed that the reorganization would make the police less responsive to citizens and would thus lead to more crime. The police have compiled theft statistics from the years following the reorganization that show that the critics were wrong. There was an overall decrease in reports of thefts of all kinds, including small thefts. **Question:** Which of the following, if true, most seriously challenges the mayor's argument? | 1. In other cities where police departments have been similarly reorganized, the numbers of reported thefts have generally risen following reorganization.

2. When city police are perceived as unresponsive, victims of theft are less likely to report thefts to the police.

3. The mayor's critics generally agree that police statistics concerning crime reports provide the most reliable available data on crime rates.

4. The mayor's reorganization of the police department failed to save as much money as it was intended to save. | 1 |

| 218 | Jupiter is a gas giant planet and the largest planet in the solar system. Its mass is 2.5 times the total mass of the other seven planets in the solar system. Observations have found that most of the more than 70 moons surrounding Jupiter are composed of water ice. Therefore, Jupiter's atmosphere should contain a considerable amount of water. **Question:** Which of the followings, if true, can best support the above statement? | 1. After hundreds of millions of years, the satellite may slowly fall onto the planet. 
 2. Many of the water in interstellar space exists in gaseous form. 
 3. Uranus is also a gas giant planet, and it has been confirmed that it contains a lot of water ice. 
 4. The satellite and the planets around it were formed from the same gas and dust at the same time. | 3 |
|---|---|---|---|
| 10376 | Lake Dali is a barrier lake on the plateau formed by volcanic eruptions. Like salmon living in the sea, Hua Zi fish-Leuciscus waleckii, which lives in a brackish lake, must migrate to the upper reaches of the Tanshui River to spawn and breed, although the four rivers currently flowing into Lake Dali are inland rivers, and none of them leads to the sea. Scientists are still convinced that the Huaziyu in Lake Dali first migrated from the ocean. **Question:** Which of the following options, if true, provides the best explanation for scientists' beliefs? | 1. The Leuciscus waleckii that lives in the waters such as Heilongjiang is twice as big as the Leuciscus waleckii fish in Lake Dari. 
 2. The caught Hua Zi fish can only survive for a day or two after being put into sea water or fresh water, and will decay quickly after death. 
 3. Melting glaciers will form Lake Dali, and the overflowing lake was once connected to the Liao River, which flowed into the ocean. 
 4. The researchers put the fry of Hua Zi fish in Dali Lake into Gainao thousands of miles away, and the culture was successful. | 2 |
| 13334 | It is repeatedly claimed that the dumping of nuclear waste poses no threat to people living nearby. If this claim could be made with certainty, there would be no reason for not locating sites in areas of dense population. But the policy of dumping nuclear waste only in the more sparsely populated regions indicates, at the very least, some misgiving about safety on the part of those responsible for policy. **Question:** Which one of the following, if true, would most seriously weaken the argument? | 1. Until there is no shred of doubt that nuclear dumps are safe, it makes sense to situate them where they pose the least threat to the public. 
 2. There are dangers associated with chemical waste, and it, too, is dumped away from areas of dense population. 
 3. In the event of an accident, it is certain that fewer people would be harmed in a sparsely populated than in a densely populated area. 
 4. Dumping of nuclear waste poses fewer economic and bureaucratic problems in sparsely populated than in densely populated areas. | 3 |

Table 9: Cross-validation evaluation: ReClor and LogiQA2.0 BoN accuracies, black numbers are the results of cross-validation. In the cross-validation evaluation, CodePMP can still bring stable improvements on different test sets.

| RM Finetune | Model | PMP | Test Set | |
| --- | --- | --- | --- | --- |
| | | | ReClor | LogiQA2.0 |
| ReClor-pair | 1.5B | ✗ | 0.534 | 0.4592 |
| | | ✓ | 0.582 | **0.4981** |
| | 7B | ✗ | 0.724 | 0.5453 |
| | | ✓ | 0.756 | **0.5835** |
| LogiQA2.0-pair | 1.5B | ✗ | 0.710 | 0.5293 |
| | | ✓ | **0.752** | 0.5504 |
| | 7B | ✗ | 0.748 | 0.6371 |
| | | ✓ | **0.794** | 0.6779 |

