# OpenReview forum: "CodePMP: Scalable Preference Model Pretraining for Large Language Model Reasoning"
_ICLR.cc/2025/Conference — Submitted to ICLR 2025_

### Official Review · Reviewer_M96b · 2024-10-28

**Soundness:** 2
**Presentation:** 3
**Contribution:** 3
**Rating:** 5
**Confidence:** 3

**Summary:**

This paper presents CodePMP, a novel pipeline for preference model pretraining aimed at enhancing the reasoning capabilities of large language models (LLMs). The primary challenge addressed is the scarcity of high-quality preference data, which is crucial for reinforcement learning from human feedback (RLHF) but costly to annotate. The authors propose a solution by leveraging code-preference pairs synthesized from publicly available high-quality source code, such as from GitHub, to improve reward model finetuning.

The CodePMP pipeline generates millions of code-preference pairs by using two different code models (a stronger model and a weaker model) to create a “chosen” and “rejected” code response for a given code prompt. These pairs are then used to pretrain the preference models, leading to better sample efficiency and improved reasoning performance in downstream tasks like mathematical reasoning (GSM8K, MATH) and logical reasoning (ReClor, LogiQA2.0).

**Strengths:**

**1. Efficiency in Reward Model Fine-Tuning**


CodePMP significantly enhances the sample efficiency of reward model (RM) fine-tuning, as the experiments indicate that fewer samples are required to achieve high accuracy. The paper shows that RMs initialized with CodePMP outperform directly fine-tuned models, even with smaller amounts of data. This efficiency makes the method appealing for real-world applications where data collection can be expensive or time-consuming.

**2. Scalable Data Generation**


One of the major strengths of CodePMP is its ability to generate large-scale preference data using code-preference pairs. By synthesizing data from publicly available source code on platforms like GitHub, the authors address the scarcity of high-quality preference data. This reduces the reliance on costly human annotation, making the pretraining process significantly more scalable and potentially more cost-effective than traditional methods relying solely on human feedback.

**3. Leveraging Code for Reasoning Tasks**

The paper introduces an innovative use of GitHub-sourced code to enhance reasoning in LLMs. By leveraging the logical structure of code, the authors create code-preference pairs, where stronger models generate "accepted" responses and weaker models generate "rejected" ones. This approach capitalizes on the abundance of publicly available code on GitHub, using it to simulate reasoning tasks. The structured nature of code makes it an ideal dataset for pretraining models to improve problem-solving and logical reasoning capabilities, aligning with recent research showing that training on code boosts reasoning performance in LLMs.

**Weaknesses:**

**1. Lack of Examples**

The paper lacks clear and detailed examples that illustrate how CodePMP actually solves reasoning tasks using code-preference pairs. While the overall methodology is explained, it's hard for the reader to grasp how reasoning is specifically improved through the use of code data. The absence of step-by-step examples of how the code-preference pairs are used to train the reward model leaves an important gap. For instance, showing a few real examples of code, its corresponding “chosen” and “rejected” responses, and how these responses contribute to solving a reasoning task would be highly valuable. The appendix, while providing some additional technical details, does not adequately fill this gap.


**2. Limited Baselines**

The paper’s comparison to **alternative methods** is quite limited. It focuses primarily on showing the improvements over direct RM fine-tuning without pretraining, but fails to benchmark CodePMP against other advanced reasoning techniques such as:

Chain-of-Thought (CoT) prompting, which has been shown to significantly improve reasoning performance by breaking down tasks into intermediate steps.
Search-based methods like Monte Carlo Tree Search (MCTS), which have been effective in enhancing reasoning capabilities in language models through external computation and verification steps.
A reader would expect a thorough comparison of CodePMP against these methods, especially since these are widely used in the current state-of-the-art LLM systems. Additionally, the paper should clarify:

How does CodePMP stack up against CoT and search-based approaches in the same experimental setup?
What are the strengths and weaknesses of CodePMP relative to these techniques?

**3. Challenges in Maintaining Data Quality with Evolving LLM Capabilities**

A key limitation of the CodePMP methodology is its reliance on two distinct models—one to generate "accepted" (high-quality) responses and another to produce "rejected" (low-quality) responses. While this setup works for synthesizing large quantities of training data, there is a valid concern that as LLMs improve, the gap between the two models could narrow, leading to mismatches in the data quality. For example, there is a risk of **data distribution drift:** As the strong model improves, the generated "accepted" responses may no longer reflect the kind of reasoning tasks the preference model was initially trained on, leading to a mismatch between the training data and the actual data seen in practice. The system might overfit to synthetic training data that does not generalize well to the actual tasks faced by stronger LLMs.

**Questions:**

Given that CodePMP relies heavily on code data, how well do you expect it to generalize to reasoning tasks that do not have a structured, logical basis like code (e.g., common-sense reasoning, social interactions, or natural language inference)? Have you explored tasks beyond mathematical and logical reasoning?

---

> ### Author Response · Authors · 2024-11-25
> **Response to Reviewer M96b**
>
> Thank you for your comments. I will address your questions one by one.
>
> **Q1**: Lack of Examples
>
> **A1**: Thank you for your suggestion. We will include more examples and corresponding analysis in future versions of the paper. In the meantime, we have open-sourced the dataset to help provide access to additional examples. However, due to anonymity requirements during the review period, we are unable to share the open-source link at this time.
>
> **Q2**: How does CodePMP stack up against CoT and search-based approaches in the same experimental setup? What are the strengths and weaknesses of CodePMP relative to these techniques?
>
> **A2**: Thank you for this insightful question! CodePMP is very suitable to integrated with CoT and search-based approaches. However, since this is a broad topic, we only mention some of our preparations, especially GenPMP and CodePrMP, in the future work section (Section 7). We would like to provide some recent progress, as follows:
> - **GenPMP** models CodePMP in a generative way. It introduces CoT and critic techniques in the context and has brought promising improvement to CodePMP.
>
> | BoN   | GSM8K (CodePMP) | GSM8K (GenPMP) | MATH (CodePMP) | MATH (GenPMP) |
> |-------|------------------|----------------|----------------|---------------|
> | N=1   | 0.7718           | 0.7718         | 0.298          | 0.298         |
> | N=4   | 0.8324           | 0.818          | 0.34           | 0.36          |
> | N=32  | 0.8234           | 0.84           | 0.348          | 0.36          |
> | N=256 | 0.837            | 0.8544         | 0.344          | 0.4           |
>
> - **CodePrMP** integrates with search-based approaches such as MCTS, which also involves process reward modeling (PRM). By pretraining on process-rewarded code, we aim to enhance PRM's effectiveness, enabling better synergy with MCTS.
>
> **Q3**: Challenges in Maintaining Data Quality with Evolving LLM Capabilities
>
> **A3**: This is indeed an important and insightful concern! The issue of "data distribution drift" can lead to reward hacking. One of the key motivations behind designing CodePMP was to improve the generalization ability of the preference model and mitigate reward hacking through sufficiently broad and diverse prompts and responses in the broadly-covered code domain.
> However, we acknowledge that continuously iterating pairwise data is essential to adapt to evolving LLM capabilities. From the perspective of scalable oversight, when model capabilities surpass human capabilities, we may need to leverage "critic of critic" methods to ensure data quality. We are actively working on this challenge, and we also hope that more peers will pay attention to our work and give us the same valuable suggestions as you.
>
> **Q4**: Generalization of CodePMP to Non-Structured Reasoning Tasks
>
> **A4**: Currently, our work primarily focuses on reasoning tasks with a structured, logical basis. However, your point is well-taken, and there is significant potential for extending this work to areas like common-sense reasoning, social interactions, and natural language inference. We are actively open-sourcing our research results to inspire and support the broader community in collaboratively advancing reasoning capabilities across diverse domains.
>
> We hope that our response addresses your concerns and strengthens your confidence in CodePMP’s contribution to the research community.

---

> > ### Author Response · Authors · 2024-11-27
> > **Looking Forward to the Response from Reviewer M96b**
> >
> > Dear Reviewer M96b,
> >
> > We sincerely appreciate the insightful feedback you provided, which has been instrumental in enhancing the quality of our work. We have prepared detailed responses for your review.
> >
> > Please let us know if we need to provide any further clarifications or feedback.
> >
> > Best regards,
> >
> > The Authors

---

> > > ### Comment · Reviewer_M96b · 2024-12-02
> > >
> > > Thank you for your response!
> > > I'm still curious about how you generate code prompts for specific problems. Could you provide several examples?

---

### Official Review · Reviewer_zxHN · 2024-11-02

**Soundness:** 2
**Presentation:** 2
**Contribution:** 2
**Rating:** 5
**Confidence:** 3

**Summary:**

This work introduce CodePMP, a pipeline that create synthesize code-preference pairs in order to have more high quality preference data. The authors show that a better reward model can be obtained with these preference pair data. And it also shows the improvements in several reasoning tasks (GSM8K, MATH, ReClor, LogiQA2.0) with these synthesize code-preference pairs data.

**Strengths:**

1. A pipeline that create synthesize code-preference pairs is introduce in this work. It can help to solve the scarcity of high-quality preference data if it is working well.

2. Large improvements are achieved on several reasoning tasks (GSM8K, MATH, ReClor, LogiQA2.0) with these synthesize code-preference pairs data.

3. Some details of CodePMP are shown. These can be helpful to the community.

**Weaknesses:**

1. The experiments details are missing or confused. It is better to clarify in the next version. Please check the following Questions section for more details.

2. In the experiments, only two Qwen2 models are used for the evaluation. Other model family results can be used for the verification of the methods.

**Questions:**

1. In the Data Construction, a description summarizer is used to generate prompts that describe the code. There is no any detail on this part. And what is quality of constructed data pair? There is no any related analysis in this work. It is not a human-annotated dataset. So it is important to show the data quality.

2. In Section 4.2.2, how the different number of N candidate responses are generated? Is Qwen2 model used to sample these candidate response? There is no details about this.

3. In Figure 4, the author claims that: RM with Code PMP initialization consistently outperform. However, it seems not right. Figure 4 (g) shown a counter part case. RM with Code PMP initialization is consistently  worse than RM without Code PMP initialization. Please check in detail.

4. As shown in Figure 5, increasing the number of code-preference pairs consistently improve accuracy. Look like there is still some improvement space. Do you think more preference data can be obtained without too much work?

Other questions: is the PMP data planed to be released? how many code-preference pairs? (only file number and token number are shown, e.g, Python files, with 20 million files and 13.1 billion tokens,)

---

> ### Author Response · Authors · 2024-11-25
> **Response to Reviewer zxHN**
>
> Thank you for your suggestions.
>
> **Q1**：In the experiments, only two Qwen2 models are used for the evaluation. Other model family results can be used for the verification of the methods.
>
> **A1**: Thank you for this suggestion. As noted in the "General Author Response," we conducted additional experiments using Llama3.2 as the generator. Moreover, we provided the verification results of the CodePMP method on the Gemma2 model in Section 5.4 and Figure 8. These results demonstrate the general applicability of CodePMP, significantly enhancing models across different families, including Gemma2 and Llama3.2.
>
> **Q2**: In the Data Construction section, a description summarizer is used to generate prompts that describe the code. However, there are no details on this part, and the quality of the constructed data pairs is not analyzed. Since this is not a human-annotated dataset, it is crucial to demonstrate the data quality.
>
> **A2**: We apologize for omitting some details regarding prompt generation and data quality.
> - **Prompt generation**: Specifically, we employed deepseekcoder-6.7b with optimized instructions to generate code prompts from code snippets. In future versions, we will include the prompts and examples in the appendix for clarity.
> - **Data quality**: To ensure the quality of automatically labeled data, we designed a critique prompt to score the label correctness of the generated pairs. On a test set of 100 pairs, we iteratively refined the critique prompt until it achieved 95% agreement between GPT4-o and experienced programmers (human-annotators). Using GPT4-o, we then scored 2,000 generated pairs, and the label accuracy reached 76%. Given that this is preference model (PM) **pretraining**, we balanced quality and diversity. We are exploring scalable and cost-effective ways to optimize synthetic data quality while reducing potential bias. Encouragingly, as highlighted in our "General Author Response" and the experimental results in the paper, CodePMP has already demonstrated strong generalizability and scalability, with promising room for further improvement in data quality.
>
> **Q3**: In Section 4.2.2, how are the different numbers of N candidate responses generated? Is the Qwen2 model used to sample these candidate responses? There are no details provided.
>
> **A3**: To generate varying numbers of N candidate responses: We used Mistral-7B as the generator, sampling each question in the GSM8K and MATH-500 datasets 256 times with a temperature of 0.7 and top-p set to 1. To ensure consistency in the Best-of-N evaluation, we selected the same N candidate responses from these 256 samples for each Reward Model (RM). These candidates were scored using the RM, and the highest-scoring response was selected as the best response for each question.
> In the primary experiments, Mistral-7B served as the generator for the Best-of-N evaluation. To explore the performance gains of CodePMP with a more advanced generator, we later used Qwen2-Math-7B-Instruct, which excels in mathematical problem-solving. Results are provided in "General Author Response", which shows that CodePMP can even bring significant improvements to Qwen2-Math-7B-Instruct.
>
> **Q4**: In Figure 4, the author claims that RM with CodePMP initialization consistently outperforms. However, Figure 4(g) seems to show the opposite case, where RM with CodePMP initialization is consistently worse. Please check this in detail.
>
> **A4**: We sincerely apologize for the confusion caused by the reversed color labels in Figure 4(g). After rechecking, Figure 4(g) still demonstrates that RM with CodePMP initialization consistently outperforms RM without CodePMP initialization. We will fix this issue in future versions. Thank you for pointing it out!
>
> **Q5**: As shown in Figure 5, increasing the number of code-preference pairs consistently improves accuracy. It seems there is still room for improvement. Do you think more preference data can be obtained without too much work?
>
> **A5**: Yes, there is substantial potential for further improvement. Open-source platforms like GitHub contain a vast amount of code that can be used to construct code preference data. We have only utilized a small fraction of this resource. We plan to open-source our data generation pipeline to benefit the entire research community.
>
> **Q6**: Is the PMP data planned to be released? How many code-preference pairs are included?
>
> **A6**: The **28 million** PMP code-preference pairs have already been released. However, due to the anonymity policy, we cannot provide the corresponding link here at this time.
>
> We hope that our response addresses your concerns and strengthens your confidence in CodePMP’s contribution to the research community.

---

> > ### Author Response · Authors · 2024-11-27
> > **Looking Forward to the Response from Reviewer zxHN**
> >
> > Dear Reviewer zxHN,
> >
> > We sincerely appreciate the insightful feedback you provided, which has been instrumental in enhancing the quality of our work. We have prepared detailed responses for your review.
> >
> > Please let us know if we need to provide any further clarifications or feedback.
> >
> > Best regards,
> >
> > The Authors

---

### Official Review · Reviewer_2FUY · 2024-11-04

**Soundness:** 2
**Presentation:** 2
**Contribution:** 2
**Rating:** 5
**Confidence:** 4

**Summary:**

The paper presents codepmp, a scalable pretraining method designed to improve the reward model by leveraging synthesized code-preference pairs. CodePMP collects a vast repository of publicly available source code from GitHub and leverages a strong and a weak model to generate synthetic preference pairs to pretrain a reward model. Utilizing this pretrained reward model, LLM's ability to solve reasoning tasks is largely improved by best of N sampling.

**Strengths:**

1. This work proposes an interesting idea of preference pre-training.
2. This method is automated, reducing the dependency on manually annotated preference data.
3. With best-of-N strategy, CodePMP could improve LLM's reasoning performance.

**Weaknesses:**

1. Unfortunately, the effectiveness of the RM was not fully verified, e.g. by using the RM for RFT or PPO training.
2. It remains unclear whether and how code training could help reasoning tasks in natural language. It would be great if the authors could have explored more on relationship between coding and reasoning tasks for model training.
3. No mention of whether the dataset will be open sourced.
4. The models used in data construction are limited. It would be helpful to verify the generalization with more number of generators, at least the ones used to construct the training data.
5. Evalution tasks are limited. Regular coding benchmarks like humaneval and mbpp should be relevant.

**Questions:**

1. Figure 4 and 5 are too small to watch.
2. Why CodePMP can help reward modeling in mathematical tasks? Please give more analysis in the paper.
3. Reference formatting issues, such as line 429.
4. Would you consider using a larger reward model to verify the scalability?

---

> ### Author Response · Authors · 2024-11-25
> **Response to Reviewer 2FUY**
>
> Thank you for your comments. Below, we provide responses to each question in detail.
>
> **Q1**: Unfortunately, the effectiveness of the RM was not fully verified, e.g., by using the RM for RFT or PPO training.
>
> **A1**: We follow prior works ([1][2][3]) that primarily use Best-of-N performance as the evaluation protocol for learned RMs. When N is sufficiently large (e.g., N=256 in our experiments), it is to approximate the performance ceiling of the current policy, making it a reliable metric to assess RM capability.
> While RFT or PPO training can also validate RM effectiveness, they are more resource-intensive as they require policy training. Although we believe Best-of-N is sufficient to evaluate the improvements CodePMP brings to RMs, incorporating RFT or PPO training in future work could enhance confidence in the method.
>
> *References*:
> - *[1]Zhang et al. (2024) "Generative Verifiers: Reward Modeling as Next-Token Prediction"*
> - *[2]Cobbe et al. (2021) "Training Verifiers to Solve Math Word Problems"*
> - *[3]Lightman et al. (2023) "Let's Verify Step by Step"*
>
> **Q2**: It remains unclear whether and how code training could help reasoning tasks in natural language. It would be great if the authors could explore the relationship between coding and reasoning tasks for model training.
>
> **A2**: CodePMP enhances the reward model (RM), which significantly improves Best-of-N (BoN) performance in mathematical and logical reasoning tasks. The stronger the RM, the closer BoN performance approaches the Generator’s capability limit. However, training a robust RM requires large-scale, high-quality labeled data, which is often scarce due to two challenges:
>
> - **Lack of prompts**: Reasoning-related prompts are hard to collect and often need to be synthesized. Code, which inherently embodies reasoning properties, provides an abundant source of high-quality data, as it is designed to solve specific tasks. Code fragments can be used to synthesize large-scale reasoning-related prompts.
> - **Difficulty in constructing preference data**: Manual labeling for reasoning tasks is inefficient and costly. However, with advancements in CodeLLMs, stronger CodeLLMs can generate better solutions for code prompts compared to weaker ones, enabling automated preference data construction.
>
> By addressing these limitations, we can scale reasoning-related prompts and preference pairs. Pretraining on these preferences provides substantial reasoning capabilities (as shown in Figure 4). Minimal RM fine-tuning further boosts performance on mathematical and logical reasoning tasks. We will include more detailed analysis in future versions.
>
> **Q3**: No mention of whether the dataset will be open-sourced.
>
> **A3**: The dataset has already been open-sourced on HuggingFace. However, due to anonymity requirements, we cannot provide the link at this time.
>
> **Q4**: It would be helpful to verify generalization with more generators.
>
> **A4**: Following your suggestion, we further used Qwen2-Math-7B-Instruct as the generator, which excels in mathematical problem-solving. The results, included in the "General Author Response," show that CodePMP brings improvements to Qwen2-Math-7B-Instruct as well.
>
> **Q5**: Figures 4 and 5 are too small to read.
>
> **A5**: We apologize for the inconvenience. Due to space limitations, we reduced the size of the figures. We will enlarge them in future versions for better readability.
>
> **Q6**: Why does CodePMP help reward modeling in mathematical tasks? Please provide more analysis in the paper.
>
> **A6**: CodePMP benefits reward modeling for mathematical tasks because of the intrinsic connection between code data and mathematical problem-solving. Code represents algorithmic implementations, which are essential for solving math problems.
> Thus, training a discriminator on code preference data improves the model's ability to accurately evaluate mathematical solutions. We will include more examples and analysis in future versions to clarify this further.
>
> **Q7**: Reference formatting issues, such as on line 429.
>
> **A7**: We apologize for this issue and will fix it immediately.
>
> **Q8**: Would you consider using a larger reward model to verify scalability?
>
> **A8**: As mentioned in the "General Author Response," we followed your suggestion and verified CodePMP’s scalability on Qwen2-72B. The results demonstrate that CodePMP significantly enhances the performance of larger, more powerful models.
>
> We hope that our response addresses your concerns and strengthens your confidence in CodePMP’s contribution to the research community.

---

> > ### Comment · Reviewer_2FUY · 2024-11-26
> >
> > Thank you for the response. One question missed:
> >
> > Q: Evalution tasks are limited. Regular coding benchmarks like humaneval and mbpp should be relevant.
> >
> > Could you please also elaborate on that?

---

> > > ### Author Response · Authors · 2024-11-27
> > > **Evalution on humaneval task.**
> > >
> > > We further validated the effectiveness of CodePMP on the code task HumanEval, and the results are shown in the table below. We used deepseek-coder-6.7b-instruct as the generator. As can be seen, the RM initialized with CodePMP shows a more significant improvement and experiences a slower performance decline.
> > >
> > > | RM      | Qwen2-7B w/o CodePMP | Qwen2-7B w/ CodePMP |
> > > |---------|-----------------------|---------------------|
> > > | Finetune Set | CodeUltraFeedback_binarized Train-Set (8.5k pairs) | CodeUltraFeedback_binarized Train-Set (8.5k pairs) |
> > > | Generator | deepseek-coder-6.7b-instruct | deepseek-coder-6.7b-instruct |
> > > | HumanEval(Pass@1, 0-shot) | HumanEval(Pass@1, 0-shot) | HumanEval(Pass@1, 0-shot) |
> > > | BoN=1   | 0.7134                | 0.7134              |
> > > | BoN=2   | 0.7317                | 0.7195              |
> > > | BoN=4   | 0.7073                | 0.7622              |
> > > | BoN=8   | 0.6890                | **0.7683**            |
> > > | BoN=16  | 0.6951                | 0.7256              |
> > > | BoN=32  | 0.6585                | 0.7378              |
> > > | BoN=64  | 0.6829                | 0.7134              |
> > > | BoN=128 | 0.6707                | 0.7012              |
> > > | BoN=256 | 0.6707                | 0.7195              |
> > >
> > > We hope that our response addresses your concerns and strengthens your confidence in CodePMP’s contribution to the code community.

---

> ### Comment · Reviewer_2FUY · 2024-12-01
>
> Thank you for your response. I am raising my rating to 5 as some of my concerns are addressed. I am not fully convinced of the effectiveness of the final RM model without a result from its application in model optimization (e.g. DPO) though. Besides, If the claim "When N is sufficiently large (e.g., N=256 in our experiments), it is to approximate the performance ceiling of the current policy" is true, the humaneval experiment shows that the ceiling of the current policy is close to Bo1 without CodePMP.
>
> I think the authors propose an interesting idea for RM pre-training though. CodePMP could be a good research step if the authors can add more convincing experiment results. I encourage the authors to further polish the work and make it a solid one.

---

### Official Review · Reviewer_YonS · 2024-11-04

**Soundness:** 2
**Presentation:** 3
**Contribution:** 2
**Rating:** 3
**Confidence:** 4

**Summary:**

The paper presents CodePMP, a novel and scalable preference model pretraining (PMP) pipeline aimed at enhancing the reasoning capabilities of Large Language Models (LLMs).

CodePMP builds its dataset in two stages: 1) generating instructions from GitHub files using a CodeLLM, and 2) generating two responses for each instruction from a strong CodeLLM and a weak CodeLLM, which are the preferred and rejected response pairs. 28 million such response pairs are constructed.

After pre-training an LLM on these 28 million pairs, CodePMP improves RM finetuning efficiency. The resulting reward model built upon CodePMP can then be leveraged to improve performance on reasoning tasks, including solving mathematical and logical problems.

**Strengths:**

- The proposed method is highly scalable, allowing it to create 28 million preferred and rejected response pairs.

- The reward model can improve two different reasoning tasks (math reasoning and logic reasoning).

**Weaknesses:**

- CodePMP creates preference pairs for coding tasks, but coding tasks are not evaluated in experiments.

- The CodePMP data is constructed by deepseek-coder-instruct. It would be interesting to see whether CodePMP can further improve deepseek-coder-instruct on coding tasks.

- The reward model is initialized using Qwen2 models (Qwen2-1.5B and Qwen2-7B), which are more capable in math reasoning than the math generator MetaMath-Mistral-7B in Section 4.1.3. A more meaningful setting would be to examine whether the Qwen2 based reward model can further improve the performance of Qwen2 or Qwen2-Math on math reasoning tasks.

- Majority voting should be included as a baseline in Figure 3.

- Further analysis is needed to understand why the coding-based CodePMP contributes to improvements in math and logical reasoning.

**Questions:**

- Is Majority voting@K comparable with reranking top K with a RM in Figure 3?

---

> ### Author Response · Authors · 2024-11-25
> **Response to Reviewer YonS**
>
> Thank you for your comments.
>
> **Q1**: CodePMP creates preference pairs for coding tasks, but coding tasks are not evaluated in experiments.
>
> **A1**:
> CodePMP is not specifically designed for coding tasks. Its main goal is to enhance mathematical reasoning and logical reasoning tasks through code preference pre-training.
> We do include performance comparisons on coding benchmarks in Section 5.5 and Table 2 (page 9). For clarity, we have listed the relevant results here. As shown, the improvement brought by PMP is significant. In the table, "w/o" denotes without PMP, and "w/" denotes with PMP.
>
> |  Qwen2- | **PMP**   | **Coding RM Accuray** |
> |----------|-------|--------|
> | 1.5b | w/o | 0.6841  |
> |   | w/ | **0.758**  |
> | 7b     | w/o | 0.6912  |
> |  | w/ | **0.7619**  |
>
> **Q2**: The CodePMP data is constructed by deepseek-coder-instruct. It would be interesting to see whether CodePMP can further improve deepseek-coder-instruct on coding tasks.
>
> **A2**:
> We used deepseek-coder-33b-instruct as the backbone of the reward model (RM). As shown in the results below, an RM trained with CodePMP (constructed using code preference pairs synthesized from deepseek-coder-6.7b-instruct and deepseek-coder-1.3b-instruct) outperforms an RM trained without CodePMP in downstream preference fine-tuning tasks. In other words,  CodePMP can further improve deepseek-coder-instruct on the coding preference task.
>
> |   | deepseek-coder-33b-instruct w/o PMP | deepseek-coder-33b-instruct w/ PMP |
> |--------------------------|--------------------------|--------------------------|
> | Finetune Set            | CodeUltraFeedback_binarized Train-Set (8.5k pairs) | CodeUltraFeedback_binarized Train-Set (8.5k pairs) |
> | Test Set                | CodeUltraFeedback_binarized Test-Set (1k pairs)   | CodeUltraFeedback_binarized Test-Set (1k pairs)   |
> | RM Acc                  | 0.8016                              | 0.8494                             |
>
> **Q3**: A more meaningful setting would be to examine whether the Qwen2-based reward model can further improve the performance of Qwen2 or Qwen2-Math on math reasoning tasks.
>
> **A3**:
> As noted in our "General Author Response," we further experiment with Qwen2-7B-Math as the generator.
> The results show that the RM initialized with PMP improves the BoN performance of Qwen2-7B-Math on GSM8K and MATH (Bo4). In contrast, RMs trained directly without CodePMP consistently lead to performance degradation, regardless of the value of N. Additionally, we observed that as the generator is already very strong, weaker RMs tend to misrank outputs (a phenomenon akin to reward hacking). CodePMP alleviates this issue and improves overall performance.
>
> **Q4**: Majority voting should be included as a baseline in Figure 3.
>
> **A4**:
> We have tested the majority voting approach on the GSM8K and MATH datasets. The results are summarized below. As seen in the table, CodePMP outperforms majority voting, with the improvement being more pronounced on the harder dataset (MATH).
>
> |  | GSM8K   | MATH |
> |----------|-------|--------|
> | **PMP** | **0.8484** |  **0.41** |
> | **Majority voting** | 0.8453| 0.37|
>
> **Q5**: Further analysis is needed to understand why the coding-based CodePMP contributes to improvements in math and logical reasoning.
>
> **A5**:
> CodePMP improves the reward model (RM), thereby boosting BoN performance of Generators in mathematical and logical reasoning.  The stronger the RM's capability, the closer the BoN performance is to the Generator's capability limit. Training a strong RM requires a large amount of carefully labeled data, and such data is scarce. The limitations are mainly twofold:
>
> - **Lack of prompts**: Reasoning-related prompts are difficult to collect directly and need to be synthesized. High-quality coding data is abundant, and code inherently possesses reasoning properties as it is designed to solve tasks. Code fragments can be used to generate large-scale code prompts, most of which are reasoning-related.
> - **Difficulty in constructing preference data**: Reasoning tasks are complex, and manual labeling is inefficient and expensive. However, with the rapid development of CodeLLMs, a reasonable assumption is that stronger CodeLLMs generate better solutions and outcomes for code prompts compared to weaker ones. This assumption enables automated preference data construction.
>
> By addressing these two limitations, we can generate scalable reasoning-related code prompts and preference pairs. After pre-training with these preferences, the model gains substantial reasoning capabilities (as seen in Figure 4, where models with PMP show strong performance even without RM fine-tuning). A small amount of RM fine-tuning then further enhances performance on math and logical reasoning tasks. We will add detailed analysis in the future version.
>
> We hope these additional experiments and responses can address your concerns and bolster your confidence in the improvements that CodePMP contributes to reasoning tasks.

---

> > ### Author Response · Authors · 2024-11-27
> > **Looking Forward to the Response from Reviewer YonS**
> >
> > Dear Reviewer YonS,
> >
> > We sincerely appreciate the insightful feedback you provided, which has been instrumental in enhancing the quality of our work. We have prepared detailed responses for your review.
> >
> > Please let us know if we need to provide any further clarifications or feedback.
> >
> > Best regards,
> >
> > The Authors

---

### Author Response · Authors · 2024-11-25
**General Author Response**

We sincerely appreciate the thorough and thoughtful feedback from all reviewers! Some reviewers expressed concerns about the generality and scalability of the CodePMP approach. We summarize these concerns into three categories:
- Using models from different families as the PMP/RM backbone.
- Employing a stronger Qwen model as the PMP/RM backbone.
- Using a more powerful Generator.
To address these concerns, we have designed a set of supplementary experiments for each category.

**1. Using different families of models as the PMP/RM backbone**
In our paper, we have already used Qwen and **Gemma2**(refer to Section 5.4 and Figure 8) as PMP/RM backbones. Here, we supplement these results by using the recently released Llama3.2 as the PMP/RM backbone. The results are shown in the table below:

- a. RM initialized with CodePMP consistently outperforms directly trained RM in reasoning tasks, demonstrating obvious performance improvements.
- b. RM initialized with CodePMP exhibits robustness when N increases, with a delayed performance drop compared to directly trained RM.

Results on Gemma2 and Llama3.2 highlight the general applicability of CodePMP, significantly enhancing models from other families as well.

| BoN  | w/o pmp (GSM8K) | **w/ pmp (GSM8K)** | w/o pmp (MATH) | **w/ pmp (MATH)** | w/o pmp (Reclor) | **w/ pmp (Reclor)** | w/o pmp (Logiqa-v2) | **w/ pmp (Logiqa-v2)** |
|---------|-----------------|-----------------|----------------|-----------------|------------------|-----------------|-----------------|-------------------|
| N=1  | 0.7718  | 0.7718  | 0.298  | 0.298  |  |  |  |  |
| N=4  | 0.7953  | 0.8218  | 0.332  | 0.35  |  0.574  |  **0.664**  |  0.5325  |  **0.5804**  |
| N=32  | 0.7612  | **0.8302**  | 0.3  | **0.354**  |  |  |  |  |
| N=256  | 0.721  | 0.8074  | 0.252  | 0.328  |  |  |  |  |

**2. Using a stronger Qwen model as the PMP/RM backbone**
To evaluate the effects of a stronger Qwen model as the PMP/RM backbone, we conducted experiments using the largest model in the Qwen2 series, Qwen2-72B. The results are summarized in the table below:

- a. RM initialized with CodePMP consistently outperforms directly trained RM across all N values in reasoning tasks.
- b. Stronger backbone models exhibit greater robustness. For example, in the MATH dataset, RM initialized with CodePMP maintains the upward trend even when N=256.

This demonstrates the scalability of CodePMP, as it enhances a larger and more powerful model.

| BoN  | w/o pmp (GSM8K) | **w/ pmp (GSM8K)** | w/o pmp (MATH) | **w/ pmp (MATH)** | w/o pmp (Reclor) | **w/ pmp (Reclor)** | w/o pmp (Logiqa-v2) | **w/ pmp (Logiqa-v2)** |
|-------|-----------------|--------------------|----------------|-------------------|------------------|-----------------|-----------------|--------------------|
|  |  |  |  |  |  |  |  |  |
| N=1  | 0.7718  | 0.7718  | 0.298  | 0.298  |  |  |  |  |
| N=4  | 0.8453  | 0.8453  | 0.424  | 0.424  |  0.894  |  **0.918**  |  0.7117  |  **0.7927**  |
| N=32  | 0.8529  | **0.8628**  | 0.488  | 0.5  |  |  |  |  |
| N=256 | 0.8249  | 0.84  | 0.506  | **0.514**  |  |  |  |  |

**3. Using a more powerful Generator**

To test whether CodePMP can improve more powerful Generators, we chose Qwen2-Math-7B, a model specifically optimized for mathematics, as the Generator. (Note: Since Reclor and Logiqa-v2 candidate responses are not generated by the Generator, we evaluate on the GSM8K and MATH datasets.) The results are as follows:

- RM initialized with CodePMP consistently outperforms directly trained RM across all N values in mathematical reasoning tasks.
This shows that CodePMP can even bring significant improvements to stronger generator optimized specifically for mathematics.

| BoN  | GSM8K w/o PMP | GSM8K **w/ PMP** | MATH w/o PMP | MATH **w/ PMP** |
|--------|---------------|--------------|--------------|-------------|
| N=1  | 0.8635  | 0.8635  | 0.698  | 0.698  |
| N=4  | 0.8544  | **0.8931**  | 0.690  | **0.724**  |
| N=32 | 0.8446  | 0.8795  | 0.643  | 0.698  |
| N=256| 0.8256  | 0.8590  | 0.614  | 0.690  |


**Summary**
Across these three supplementary experiments, CodePMP demonstrates excellent scalability and generality. It provides consistent improvements in reasoning tasks for models of varying capabilities and sizes. We will include the detailed experiment setups in future versions of the paper.
We hope these additional experiments, combined with the extensive experiments (especially the resource-intensive extended experiments) in the paper, effectively address the reviewers' concerns. We also hope that CodePMP will provide new impetus for RLHF research and inspire our peers from a scalable perspective.

---

### Meta-Review · Area_Chair_VmS6 · 2024-12-20

**Metareview:**

This paper introduces CodePMP, a scalable preference model pretraining pipeline that leverages synthesized code-preference pairs from publicly available, high-quality source code to improve reward model fine-tuning. I appreciate the authors added additional results during the rebuttal period. However, there are some concerns still in the current work: (1)  the evaluation does not demonstrate the results of reward models in model optimization (e.g. DPO), which should be the end goal; (2) Scalability Concerns: While CodePMP is designed to be scalable, LLM models often evolve and would need continuous iteration of preference data to adapt to new LLM capabilities.

**Additional Comments On Reviewer Discussion:**

The main points raised by the reviewers:
- Evaluation Gaps: Limited evaluation to demonstrate the benefits of the proposed method on model optimization (DPO or PPO).
- Data Quality: Concerns about the accuracy and quality of synthesized preference pairs without human validation.
- Scalability with Model Evolution: Potential mismatch as LLMs improve, leading to data drift and reduced preference model generalization.

Despite the authors addressing some concerns through additional experiments during the rebuttal, key gaps remain in evaluation methodologies and the scalability of the reward model.

---

### Decision · Program_Chairs · 2025-01-22

Reject